# FEDERATED LEARNING WITH A SINGLE SHARED IMAGE

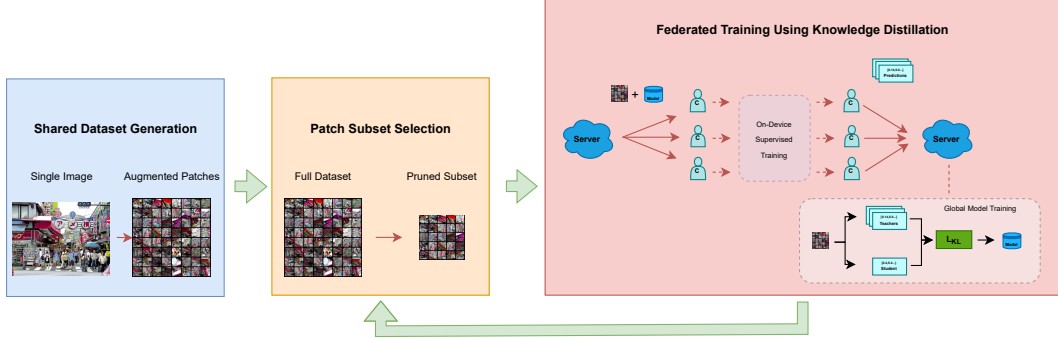

Figure 1: Schematic illustration of our federated learning algorithm using single images. Our algorithm works on the principle of generating a common distillation dataset from only one shared single image using deterministic augmentations. To this end, our method dynamically selects the best patches for the training of the global model in the next round using knowledge distillation.

## ABSTRACT

Federated Learning (FL) enables multiple machines to collaboratively train a machine learning model without sharing of private training data. Yet, especially for heterogeneous models, a key bottleneck remains the transfer of knowledge gained from each client model with the server. One popular method, FedDF, uses distillation to tackle this task with the use of a common, shared dataset on which predictions are exchanged. However, in many contexts such a dataset might be difficult to acquire due to privacy and the clients might not allow for storage of a large shared dataset. To this end, in this paper, we introduce a new method that improves this knowledge distillation method to only rely on a *single* shared image between clients and server. In particular, we propose a novel adaptive dataset pruning algorithm that selects the most informative crops generated from only a single image. With this, we show that federated learning with distillation under a limited shared dataset budget works better by using a single image compared to multiple individual ones. Finally, we extend our approach to allow for training heterogeneous client architectures by incorporating a non-uniform distillation schedule and client-model-mirroring on the server-side.

## 1 INTRODUCTION

Federated Learning (FL) is a paradigm in the field of distributed machine learning which enables multiple clients to collaboratively train powerful predictive models without the need of centralising the training data (Zhang et al., 2021). It comes with its own set of key challenges in terms of skewed non-IID distribution of data between the participating clients (Zhu et al., 2021a; Li et al., 2020; Chai et al., 2019; Hsu et al., 2019; Lin et al., 2020) and communication efficiency during training (Konečný et al., 2016; Lin et al., 2020) among others. These challenges are not directly answered with the classical approaches such as FedAvg (McMahan et al., 2023), which rely primarily on a naive client network parameter sharing approach. Since the inclusion of clients with different data

distributions has a factor of heterogeneity involved (Zhu et al., 2021a; Hsu et al., 2019), another well-known work (Li et al., 2020) counteracts this heterogeneity directly during the client training. This tries to solve one challenge related to non-iidness in private data distribution, but other key challenges related to network parameter sharing remain including concerns with privacy leakage during parameter sharing (Wang et al., 2019; Sun et al., 2021), heterogeneity of client architectures (Lin et al., 2020; Chai et al., 2019) and high bandwidth cost of parameter sharing (Konečný et al., 2016). To this end, along a second line of thought implementing a server-side training regime, approaches suggested in (Lin et al., 2020; Li & Wang, 2019; Zhu et al., 2021b; Sui et al., 2020) make use of the process of knowledge distillation (KD) (Hinton et al., 2015; Gou et al., 2021) to overcome these challenges without the exclusive need of network parameter sharing. To facilitate central network training with the help of KD, the sharing of public data is needed between the clients and the server.

In this work, we propose a novel approach of making use of a single datum source to act as the shared distillation dataset in ensembled distillation-based federated learning strategies. Our approach makes use of a novel adaptive dataset pruning algorithm on top of generating the distillation data from a single source image during the training. This combination of shared data generation and instance selection process not only allows us to train the central model effectively but also outperforms the other approaches which make use of multiple small-sized images in place of a single image under a limited shared dataset budget. The use of a single datum source has added benefits in domains, where publicly available data and client resources (e.g., network bandwidth and connectivity) are limited in nature. The use of a single datum source has been explored (Asano et al., 2020; Asano & Saeed, 2023) under the settings of self-supervised learning and understanding extrapolation capabilities of neural networks with knowledge distillation, but it has not yet been explored in federated setting for facilitating model training.

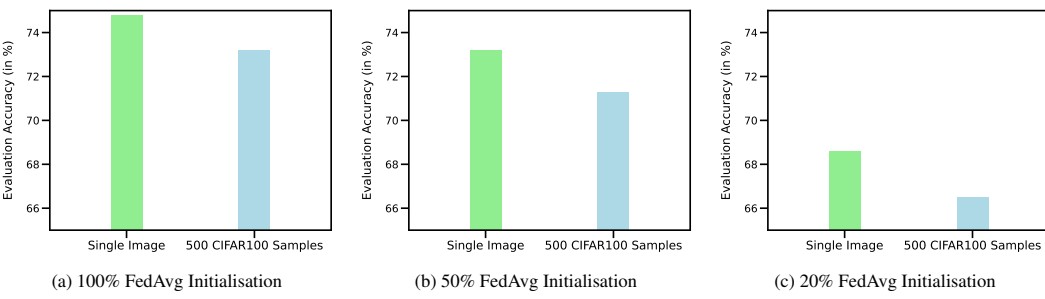

(a) 100% FedAvg Initialisation          (b) 50% FedAvg Initialisation          (c) 20% FedAvg Initialisation

Figure 2: Comparison of test performance in federated setting using a single image with patch selection compared to the equivalent size of multiple independent training samples from a labelled dataset as shared distillation dataset. We use different rates of FedAvg. initialisations to emulate different network bandwidth conditions. Detailed result in Table 4.

We perform a series of experiments to examine the viability of our proposed algorithm under varying conditions of heterogeneity in private client data, client-server model architectures, rate of pre-training network initialisations before distillation, shared dataset storage budget and real-world domain of the single images. We also extend our experiments to a use-case of heterogeneous client architectures involved during a single federated training with the help of client-model mirroring on the server side. To facilitate this, we keep one copy of the client model of each type on the server end, which acts as a global model for the clients that have the same network architecture. The global models are improved with knowledge distillation after each round of local client training with the help of shared logits over the single image dataset. The results we obtain during the aforementioned experiments demonstrate positive reinforcement towards reaching our goal of efficient federated training using Knowledge Distillation under a limited shared dataset budget.

The primary contributions of this work are :

1. Demonstrating the efficiency of a single image as a powerful medium for knowledge transfer in a federated learning setting using knowledge distillation.
2. Novel algorithm for dynamic data pruning which evolves with the current global model during federated learning.

3. Extensive evaluation of our proposed methods under a variety of conditions in a federated setting.

## 2 RELATED WORK

**Federated Learning using Knowledge Distillation**   Knowledge Distillation (KD) (Hinton et al., 2015) has been shown to successfully transfer the knowledge of an ensemble of neural networks into a single network with the means of output logits over a shared dataset. KD has also been leveraged in federated setting, such as Federated Distillation Fusion (FedDF) (Lin et al., 2020) and Federated Ensemble Distillation (FedED) (Sui et al., 2020), where the respective authors make use of KD to allow robust and faster convergence on top of using other ensembling methods such as the ones suggested in Federated Averaging (McMahan et al., 2023) for initialising the central network before the distillation training of the global model. On the other hand, authors of works such as Federated Model Distillation (FedMD) (Li & Wang, 2019) have also successfully shown that KD can be used for knowledge transfer in a federated setting for the purpose of client model personalisation. However, the application of algorithms such as FedMD is targetted for personalisation by client-side knowledge distillation rather than improvement of a central model, hence we have not delved into it in the scope of our research. In the case of ensembling methods, it has been shown in (Lin et al., 2020) that in the absence of an ensemble of local parameters before distillation training, the final test performance of the central network tends to suffer. As a result, these methods have been shown by the authors to significantly rely on parameter exchange every round similar to naive parameter exchange-based algorithms such as FedAvg (McMahan et al., 2023) for robust performance on top of KD. Since the aforementioned KD-based federated algorithms also require significant regular ensembling using network parameter exchange, our approach focusses on improving this aspect by relying significantly on knowledge distillation with the help of data pruning and augmentations on the shared public dataset, which has not yet been explored in these works.

**Communication Efficient Federated Learning**   To solve the high bandwidth costs related to parameter sharing, authors of (Caldas et al., 2018; Konečnỳ et al., 2016) have shown that quantisation of network parameters before the transfer can significantly reduce the bandwidth costs incurred during their transfer. However, with the application of the same low-bit quantization methods with the KD-based federated learning methods in (Lin et al., 2020), the authors have also shown a significant decrease in the overall test performance of models compared to their non-quantized counterparts. On the other hand to not rely on public data sources, authors of the work (Zhu et al., 2021b) have successfully shown that data-free approaches using a centrally trained generative network for producing the public shared dataset works robustly. However, this also requires an additional exchange of the generative network parameters before each round, which leads to an increase in the network bandwidth usage itself. In pursuit of reducing the bandwidth costs pertaining to network parameter exchanges as well as shared dataset sharing, these works have not yet made an attempt to make use of a storage-efficient single data source, which can simultaneously generate a public distillation dataset alongside being used for dynamic selection without added bandwidth costs. We explore this in our work.

**Single Image Representation Learning**   In (Asano et al., 2020), the authors have successfully made use of a single image to produce augmented patches for facilitating self-supervised learning of neural networks required for solving various downstream tasks. However, the focus of our work is not on the process of solving tasks with the help of self-supervised learning, but on the implications of making use of the single image patches in a federated setting as a medium of knowledge transfer for training robust classifiers. To this end, in a closely resembling work to our target task, the authors in (Asano & Saeed, 2023) have shown to be able to successfully use KD with single image patches to transfer the knowledge between a pre-trained network and an untrained network to solve the classification task of ImageNet-1k. However, the experiments by the authors were all based in a non-federated setting. In our work, we explore the usage of single image patches in a federated setting as the shared public distillation dataset and its implications in limited shared dataset budget settings.

## 3 METHOD

Our method focusses on a dynamic procedure to utilize a single image to act as a proxy dataset for distillation in addition to a federated learning setup which is similar to existing ensemble-based knowledge distillation-based methods such as FedDF (Lin et al., 2020). Alongside the generation of a distillation dataset from a single data source, we take care of dynamically selecting the best patches every round to improve the training. The two important parts of our federated strategy are described in the following sections: 3.1 **Distillation Set Generation** and 3.2 **Patch Subset Selection**.

### 3.1 DISTILLATION DATASET GENERATION

For generating meaningful image representations out of a single image, we make use of the *patchification* technique. Using this technique, we generate a large number of small-size crops from a big image by making combined use of augmentations such as 35-degree rotations, random horizontal flipping, color transformations etc, similar to the ones used in (Asano & Saeed, 2023) for knowledge distillation based learning. The image generation procedure can be controlled by a seed, which allows all the clients to be able to generate the same set of patches using the same augmentations from a single image. This provides us with the means of reducing the bandwidth usage pertaining to the transfer of the distillation proxy set to the clients required for improving the global model using Knowledge Distillation. Due to the flexibility provided by augmentations in combination with the subset selection procedure described in Section 3.2, one can make use of a single image to produce varying desired number of patches for the fixed amount of single image data.

### 3.2 PATCH SUBSET SELECTION

After we have an initial dataset for distillation using the method described in Section 3.1, we apply dataset pruning methods on this dataset to ensure the selection of information-rich patches for assisting the current round of federated training. The dataset generation procedure is based on the whole image, due to which it has the ability to produce bad representation patches such as: containing no entities, overlapping with others, being dissimilar to the target domain, and similar problems arising due to heavy augmentations and information less regions of the single image. To prune the bad patches, we make use of the following two mechanisms: **KMeans Balancing** (3) and **Entropy Selection** (3.2). These mechanisms depend on the current global model for their operation, which makes them dynamic in nature and improves their data-pruning ability with the improvement in the global model. As a result, better representations are selected with better global models.

**Entropy Selection** Entropy Selection is based on the use of randomness present in the output logits of the distillation training examples to prune their dataset. To achieve this, we examine the maximum softmax values of the logits obtained for each distillation training example using the current global model. On the basis of a removal heuristic $H^E \in \{$Top, Bottom, Random$\}$, we remove the top k percent of examples from each group (grouped on the basis of their predicted class using current global model). Top removes training examples with high softmax values while Bottom removes the ones with low softmax values. The algorithm has been described in detail in Alg. 1.

**KMeans Balancing** KMeans Balancing is based on the use of unsupervised KMeans clustering on the embedding layer representations of the training examples. To accomplish this, we establish a KMeans clustering model (based

---

**Algorithm 1:** Entropy Selection

**Input:** Distillation Training Dataset ($X$), Current Global Model ($M^G$)

**Parameters:** Percentage of Examples to Prune ($k$), Removal Heuristic ($H^E$)

**Output:** Pruned Distillation Training Dataset with Entropy Selection ($X^E$)

**begin**

  1  For all
    $x_n \in X : n \in [1..S]$ where S = size of X,
    find $Y = \{y_n : y_n =$
    Max(Softmax (Classifier Output ($M^G, x_n$)))$\}$.

  2  Select indices of the training examples
    using the removal heuristic
    $H^E \in \{$Top,Bottom,Random$\}$ with their
    corresponding values in $Y$.

  3  Push the selected indices in the new dataset
    ($X^E$).

**end**

---

on Euclidean distance) with K cluster centers and try to fit the embedding representations (using the current global model) of the distillation training examples on it. Using a selection heuristic $H^K \in \{\text{Easy, Hard, Mixed}\}$ on their calculated cluster distances $D$, we can select the training examples for the next round of training. Easy prefers examples with low cluster distance values while Hard prefers high cluster distance values. A class balancing factor $F^K \in [0.0, 1.0]$ ensures that there's a fixed lower bound for selecting a minimum number of training examples from each of the predicted classes (using the current global model) on the distillation training set. The algorithm has been described in detail in Alg. 2.

---

**Algorithm 2:** K-Means Balancing

**Input:** Distillation Training Dataset ($X$), Current Global Model ($M^G$)
**Parameters:** Number of Clusters ($K$), Size of New Dataset ($s$), Balancing Factor ($F^K$), Selection
        Heuristic ($H^K$)
**Output:** Pruned Distillation Training Dataset with KMeans Selection ($X^K$)

**begin**

1    For all $x_n \in X : n \in [1..S]$ where S = size of X, find
      $Z = \{z_n : z_n = \text{Embedding Representation } (M^G, x_n)\}$ and
      $Y = \{y_n : y_n = \text{Max-Index (Classifier Output } (M^G, x_n)\}$.

2    Define $C^P = \{\text{Set of unique classes in Y}\}$ and Number of unique classes $C = |C^P|$.

3    Initialise an independent unsupervised KMeans Clustering Model ($M^C$) using $K$ number of
      cluster centers. Fit $M^C$ on $Z$ and find $D = \{d_n : d_n = \{\text{Shortest euclidean distance of } z_n \text{ to its}$
      cluster center$\}$.

4    Define the minimum number of examples (balancing lower bound) to be selected from each class
      $c_i \in C^P$, as $LB = \lceil \frac{s}{C} * F^K \rceil$.

5    **forall** $c_i \in C^P : i \in [1..C]$ **do**
        Find the indices of examples belonging to the $c_i$ using $y_n \in Y : y_n = c_i$.
        Select indices of the new training examples on the basis of selection heuristic
         $H^K \in \{\text{Easy,Hard,Mixed}\}$ with their corresponding cluster distance values $D$.
        Push the training examples from $X$ with selected indices in the new dataset ($X^K$).
        Remove the selected training examples from $X$ and $D$.
      **end**

6    Remaining number of examples to be selected can be calculated as given by : $s - \text{size of } X^K$.

7    Using selection heuristic = $H^K$ on the cluster distance values in $D$, find the indices of the
      remaining examples to be selected. Push the training examples with selected indices in the new
      dataset ($X^K$)

**end**

---

# 4 EXPERIMENTS

## 4.1 EXPERIMENTAL SETUP

Our experimental setup for federated training using our algorithm has been shown schematically in Fig. 1.

**Dataset** We do our experiments across the following publically available datasets: CIFAR10/100 (Krizhevsky et al., 2009) and MedMNIST (PathMNIST) (Yang et al., 2023). For the distribution of private data among the clients from the collective training data, we use a similar strategy to the one suggested in (Hsu et al., 2019), which allows us to control the degree of heterogeneity using the parameter $\alpha$ (lower $\alpha$ = higher degree of non-iidness and vice-versa). We use the full test sets corresponding to the private client datasets as evaluation sets for the global model (testing is only server side). 10% of the training examples are held as a validation dataset.

For the shared public dataset, we generate patches out of a single image for all the experiments with our method. For the FedDF experiments, we have made use of CIFAR100 training set for CIFAR10 experiments, unless mentioned otherwise. The single images have been visualised in Appendix A alongside the patches and $t$-SNE visualisations during training.

**Server-Client Model Architecture**   ResNets (trained from scratch) have been used for most of our experiments as the model of choice for the server and clients (He et al., 2016). WideResNets have also been used for some of the experiments (Zagoruyko & Komodakis, 2016). The models have been explicitly defined in the table descriptions for unambiguity.

**Hyper-parameter Configuration**   The values of the learning rate (local and global) have been motivated by the experiments described in Appendix C. We use a client learning rate of 0.01 for ResNet and WResNet, while the distillation training learning rate is 0.005. For KMeans Balancing, we use a KMeans model with 1000 clusters, a class balancing factor of 1.0, and the 'Hard' selection heuristic. For Entropy selection, we remove 90% of the training examples using the 'Top' removal heuristic (Appendix B). For the experiment in Table 2, we do local client training for 10 epochs and server-side distillation for 250 steps, while 40 epochs and 500 distillation steps have been our choice for other experiments unless mentioned otherwise. We prefer to keep the use of FedAvg initialisations to 20% in our experiments unless mentioned otherwise. For all the experiments, we simulate 20 private clients, with a selection probability (C) of 0.4 per training round.

## 4.2   SELECTING THE BEST IMAGE FOR DOMAIN OF TASK

We conduct cross-dataset single-image testing using our algorithm across 3 private training datasets and 3 images, with two of them corresponding to one of the dataset domains and the third one being a random noise. The results in Table 1 exhibit that it is necessary to use a single image that is similar to the domain of the target task for optimal performance. In the case of using a single random noise image as the distillation proxy, we get the lowest test performance as it is hard for random augmentations to convert random noise patches into a knowledge transfer medium. Hence, care must be taken in choosing a single image with similar patch representations as the target task for optimal learning with our algorithm. There can be an interesting area to explore with more augmentation experiments and generative algorithms, if it is possible to use a random noise image viably as a single image with our method. We leave this as future work.

| Image | Dataset | | |
|---|---|---|---|
| | CIFAR10 | CIFAR100 | PathMNIST |
| City Street | 75.3 | 32.0 | 69.7 |
| Multi Colon Pathology Samples | 69.0 | 12.0 | 71.6 |
| Random Noise | 39.4 | 6.8 | 33.0 |

Table 1: Best test performance during 30 rounds of training using our federated method with varying Pvt. Datasets (Distribuition $\alpha = 100.0$) and 5k Single Image Patches (Distillation Proxy Set) on ResNet-8 architecture with 20% rate of FedAvg. initialisation.

## 4.3   ABLATION STUDIES WITH PATCH SELECTION MECHANISM

**Finding the Best Patch Subselection Strategies across Varying Pvt. Dataset**   To find the effectiveness of patch subset selection mechanisms, we test it under different private datasets from different real-world domains (General and Medical). Through Table 2, it is evident that the single image patches work best in the presence of a selection strategy in our federated algorithm. On their own, both KMeans Balancing (3) and Entropy Selection (3.2) strategy works better than employing no selection for the same number of patches. Together, they perform best across all the datasets we have tested which is what we use in our other experiments in this work. Both of the selection strategies and their combination significantly impact the final performance. We have done our primitive analysis with them in light of this work to find an optimal setting (Appendix B, but there might be a correlation between their settings which we have not delved into. We can propose this detailed analysis of their combinative work as future work for improving the test performance of our federated strategy with the means of better data pruning.

Through the T-SNE visualisation in Fig. 3 during different phases of federated training with a single image and our data pruning method, we observe the formation of identifiable boundary structures among the selected patches as the global model accuracy improves. This provides a visual qualitative

| Selection Strategy | Private Dataset | | |
|---|---|---|---|
| | CIFAR10 | CIFAR100 | PathMNIST |
| No Selection | $63.4 \pm 1.4$ | $24.2 \pm 1.1$ | $64.5 \pm 4.7$ |
| KMeans | $66.2 \pm 0.8$ | $21.8 \pm 2.1$ | $67.9 \pm 8.4$ |
| Entropy | $65.9 \pm 1.0$ | $26.3 \pm 1.0$ | $76.4 \pm 2.8$ |
| **KMeans + Entropy** | $67.0 \pm 1.1$ | $26.4 \pm 1.2$ | $77.1 \pm 3.0$ |

Table 2: Best test performance achieved during 30 rounds of training with different selection mechanisms (Distillation Set Size = 5000 patches) across different private datasets ($\alpha = 1.0$) using our federated strategy with ResNet-8 while using 20% rate of FedAvg. initialisations. (2 seeds)

assessment of our design claims regarding the positive correlation of the effectiveness of our patch subset selection algorithm with the evolution of the global model.

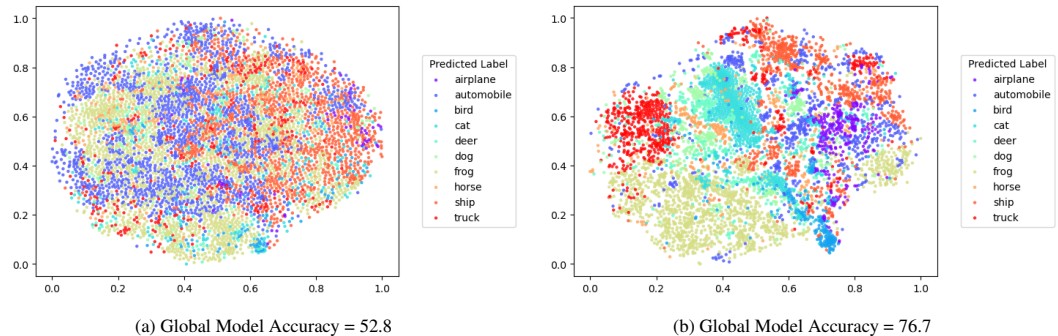

(a) Global Model Accuracy = 52.8        (b) Global Model Accuracy = 76.7

Figure 3: Scatter plot of TSNE embeddings of single image patches during different phases of training, using our method with FedAvg and ResNet-8 on CIFAR10.

**Testing the Impact of Selection Mechanism with Manually Labelled Distillation Set** We test the viability of our selection mechanism in case of extending it to the use cases where we already have a shared public dataset at hand in Table 3. During the regular exchange of initialisation parameters, the application of our selection mechanism exhibits no advantage. However, when we reduce the exchange of initialisation parameters to emulate low bandwidth conditions, it shows significant gains. This shows that even with the availability of standard distillation sets at hand in ensembled distillation methods, the subset selection mechanism can play an important role in low bandwidth cost federated training.

| Selection Mechanism Applied | FedAvg Initialisation Rate (in %) | | |
|---|---|---|---|
| | 100 | 50 | 20 |
| ✗ | $75.0 \pm 0.5$ | $73.1 \pm 1.1$ | $67.4 \pm 0.5$ |
| ✓ | $73.8 \pm 1.9$ | $72.3 \pm 0.6$ | $70.7 \pm 1.2$ |

Table 3: Comparison of best test performance during 30 rounds of training with CIFAR10 Pvt. Data with Distribuition $\alpha = 1.0$ using FedDF (with ResNet-8) between use/non-use of selection mechanism across varying rate of using FedAvg initialisation. 1000 samples from CIFAR100 train split make the distillation proxy dataset.

## 4.4 Ablation Studies with Varying Network and Storage Conditions

**Comparing Performance of a Single Image in Limited Shared Dataset Budget Settings** This is our most significant experiment in terms of exhibiting the viability of federated learning under limited shared dataset budget settings using a single image. Going through the results in Table 4, we see that for the same amount of storage budget, a single image with patch selection outperforms similarly sized individual samples. If we also lower the network budget and the rate of exchange of

initialisation parameters, it is able to hold at par with individual training samples 10 times its size. This shows a promising future for our work in the scenario where there is limited availability of publicly shared datasets as well as storage budget would be low on participating clients.

| Distillation Dataset | No. of Pixels | FedAvg Initialisation Rate (in %) | | |
|---|---|---|---|---|
| | | 100 | 50 | 20 |
| 5K CIF100 Samples | 5M | $76.4 \pm 1.4$ | $74.1 \pm 1.6$ | $68.9 \pm 1.4$ |
| **Single Image with Patch Selection** | 0.5M | $74.8 \pm 2.6$ | $73.2 \pm 3.2$ | $68.6 \pm 0.8$ |
| 500 CIF100 Samples | 0.5M | $73.2 \pm 1.7$ | $71.3 \pm 2.0$ | $66.5 \pm 0.9$ |

Table 4: Best test performance during 30 rounds of training with CIFAR10 Pvt. Data with Distribuition $\alpha = 1.0$ using ResNet-8 with different distillation datasets and rate of using FedAvg initialisation.

**Testing Performance in Limited Network Bandwidth Settings against Heterogeneous Data Distributions**  To test the impact of high data distribution heterogeneity on our FL strategy against an existing SOTA federated learning strategy based on knowledge distillation, we show the performance gains in Table 5. We also vary the network initialisation rate to test our method in high and low-bandwidth situations. We notice that with the help of patch subset selection, our methods outperform the fed strategy which doesn't make use of this process. This trend is constant across all bandwidth scenarios and local client training expenditures. We have also extended our approach to incorporate FedProx local client training regime, which shows better results than naive local client training. This extendability makes our method unique and viable to more approaches than just one kind of local training which can have added performance benefits with our algorithm.

| Strategy | Local Epochs | FedAvg Initialisation Rate (in %) | | | | | |
|---|---|---|---|---|---|---|---|
| | | 100 | | 50 | | 20 | |
| | | $\alpha = 1.0$ | $\alpha = 0.1$ | $\alpha = 1.0$ | $\alpha = 0.1$ | $\alpha = 1.0$ | $\alpha = 0.1$ |
| FedDF | 20 | $75.7 \pm 1.2$ | $48.2 \pm 2.6$ | $73.9 \pm 0.8$ | $47.3 \pm 5.2$ | $71.1 \pm 0.5$ | $42.2 \pm 9.4$ |
| | 40 | $75.7 \pm 0.9$ | $49.5 \pm 3.1$ | $74.9 \pm 1.9$ | $49.3 \pm 1.1$ | $72.5 \pm 0.5$ | $46.1 \pm 6.6$ |
| Ours w/ FedAvg | 20 | $76.9 \pm 0.6$ | $47.8 \pm 5.3$ | $75.8 \pm 0.3$ | $47.3 \pm 5.5$ | $73.7 \pm 1.0$ | $45.5 \pm 5.1$ |
| | 40 | $77.0 \pm 0.6$ | $47.8 \pm 5.4$ | $76.2 \pm 1.4$ | $49.5 \pm 2.2$ | $74.3 \pm 0.6$ | $46.6 \pm 6.7$ |
| **Ours w/ FedProx** | 20 | $77.2 \pm 0.8$ | $47.2 \pm 7.0$ | $74.5 \pm 1.3$ | $44.6 \pm 7.9$ | $73.1 \pm 0.2$ | $46.9 \pm 4.5$ |
| | 40 | $77.7 \pm 0.8$ | $47.7 \pm 3.8$ | $76.3 \pm 0.4$ | $46.0 \pm 5.3$ | $74.3 \pm 1.1$ | $45.1 \pm 6.0$ |

Table 5: Comparison of best test performance under different settings (FedAvg Initialisation Rate, Degree of Heterogenity ($\alpha$), Local Training Epochs) using different federated learning strategies with ResNet-8 on CIFAR10 during 30 rounds of training (2 seeds). 5000 single image patches have been used as distillation proxy set (w/o selection mechanism for FedDF).

## 4.5 ABLATION STUDIES WITH VARYING CLIENT-SERVER NEURAL NETWORK ARCHITECTURES

**Testing our Strategy under Homogeneous Network Architecture Settings**  We perform all the experiments in the earlier sections using ResNet-8 as the client and server models. To make sure our federated strategy works equally well among other homogenous network distributions, we put it to the test against FedDF using ResNet-20 as well as W-ResNet-16-4 in Table 6. We see that under the same distillation set storage budget, our method works better under all the tested network architectures. As per nominal expectations, network architectures with more parameters show better results than the ones with less number of parameters which enables us to achieve better test performance with more complex networks. Irrespective of the network architecture, the trend is constant when it comes to our FL strategy outperforming other strategies making use of a labelled distillation dataset in a limited storage budget scenario.

**Testing our Strategy under Heterogeneous Network Architecture Settings**  In the final experimental section, we test our federated strategy in the presence of heterogeneity in the client model

| Fed Strategy | Network Architecture | | |
|---|---|---|---|
| | ResNet-8 | ResNet-20 | W-ResNet-16-4 |
| FedDF | $67.3 \pm 1.9$ | $73.0 \pm 0.6$ | $75.3 \pm 1.2$ |
| **Ours** | $70.2 \pm 0.8$ | $74.1 \pm 0.9$ | $75.7 \pm 0.9$ |

Table 6: Best test performance during 30 rounds of training using CIF10 Pvt. Data with Distribuition $\alpha = 1.0$ using different Fed strategies and homogeneous client-server network architectures with 20% rate of FedAvg. initialisation. FedDF uses 500 CIF100 samples as distillation proxy, while our method makes use of a single image of equivalent size with patch subset selection.

architectures. The results present in Table 7 show the success of our method in training the global models when pitted against a strategy not utilising a single image. It also exhibits the importance of constant distillation training for the success of our methods, as our non-uniform approach gives subpar results with less training time. However, when going from 15k to 11k steps, we also save about 1/3 of the training time and computation resources used on the server side. It can be an interesting point of extension to our work to improve upon this non-uniform scheduling to allow for more robust training of heterogeneous models with less computation time.

| Fed Strategy | Total Distillation Steps | Macro-Avg Accuracy (Server Models) |
|---|---|---|
| FedDF | 15K | $67.4 \pm 0.6$ |
| **Ours** | 15K | $68.5 \pm 1.1$ |
| Ours w/ Scheduling | 11.3K | $65.2 \pm 1.3$ |

Table 7: Best test performance across during 30 rounds of training using CIF10 Pvt. Data with Distribuition $\alpha = 1.0$ using different Fed strategies and distillation step scheduling, under a heterogenous client distibuition (6 ResNet-8, 7 ResNet-20, 7 W-ResNet-16-4) with 20% rate of FedAvg. Initialisation. 500 CIF100 samples have been used as distillation proxy for FedDF, while our method makes use of a Single Image of equivalent size with patch selection.

# 5 CONCLUSION

Through this work, we present a novel approach for federated learning using ensembled knowledge distillation with the use of augmented image patches from a single image with patch subset selection. We successfully exhibit the performance gains with our approach in a limited shared dataset budget scenario as well as low network bandwidth requiring scenarios with less exchange of network parameters. Alongside low resource usage, the use of a single image also enables our federated strategy to be applicable to scenarios where we have a lack of public datasets required during federated training of multiple clients.

**Prospective Future of our Work** We mention a few specialised avenues of extension to our work during the discussion of results in Section 4. Some of the key points that were not mentioned in it include: Application of the single datum based federated learning to other modalities and machine learning tasks; Application of our work to other knowledge distillation-based algorithms in federated learning other than ensembled methods, such as FedMD (Li & Wang, 2019); Analysis of different kind of augmentations to improve the robustness of our method. With the aforementioned points, significant work can be done to improve the viability of our novel approach presented in this work to incorporate more real-world challenges.

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
