# Federated Learning with a Single Shared Image

# Appendix

# Table of Contents

## A  IMAGE AND PATCH VISUALISATIONS

### A.1  VISUALISATION OF SINGLE IMAGES

We make use of the images depicted in Fig. 4 as the sources for generating our distillation dataset. Kindly note, we only used the images for non-profit educational research purposes and we do not hold any rights over their commercial use. These images have been selected in correspondence to the domains of datasets we have tested in our work.

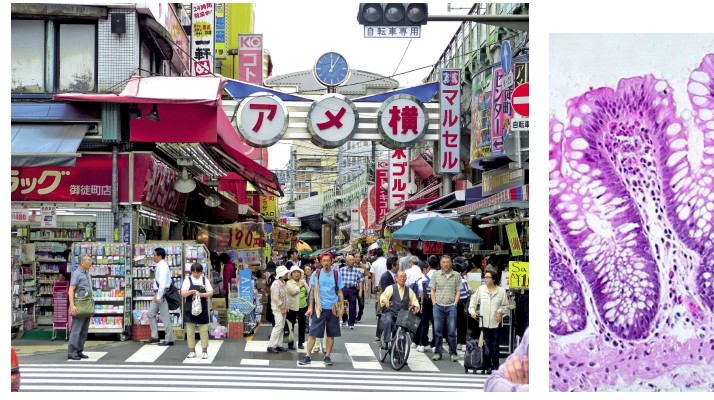
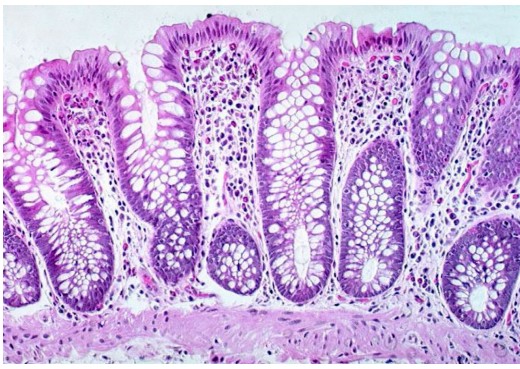

(a) A busy street in a city.          (b) Multiple colon pathology samples.

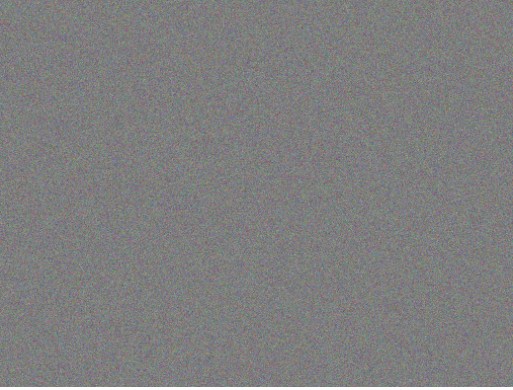

(c) Random noise.

Figure 4: Single Image sources used for our experiments for distillation dataset generation.

### A.2  VISUALISATION OF IMAGE REPRESENTATIONS FROM SINGLE IMAGES

In Fig. 3, we visualise the TSNE embeddings of the training examples in our single image distillation dataset during different phases of training on a scatterplot. It's implications have been discussed in the main paper in brief detail. To add to the earlier analysis, by looking at the difference between Fig. 3a and Fig. 3b, one can see how the embedding layer in the neural network starts forming particular regions belonging to certain classes with the increase in classification accuracy. Even though the image representations do not contain many of the classes in the classifier as is prevalent from visual inspection of Fig. 5, the pseudo labels are powerful enough to enable the learning through knowledge distillation in a federated setting.

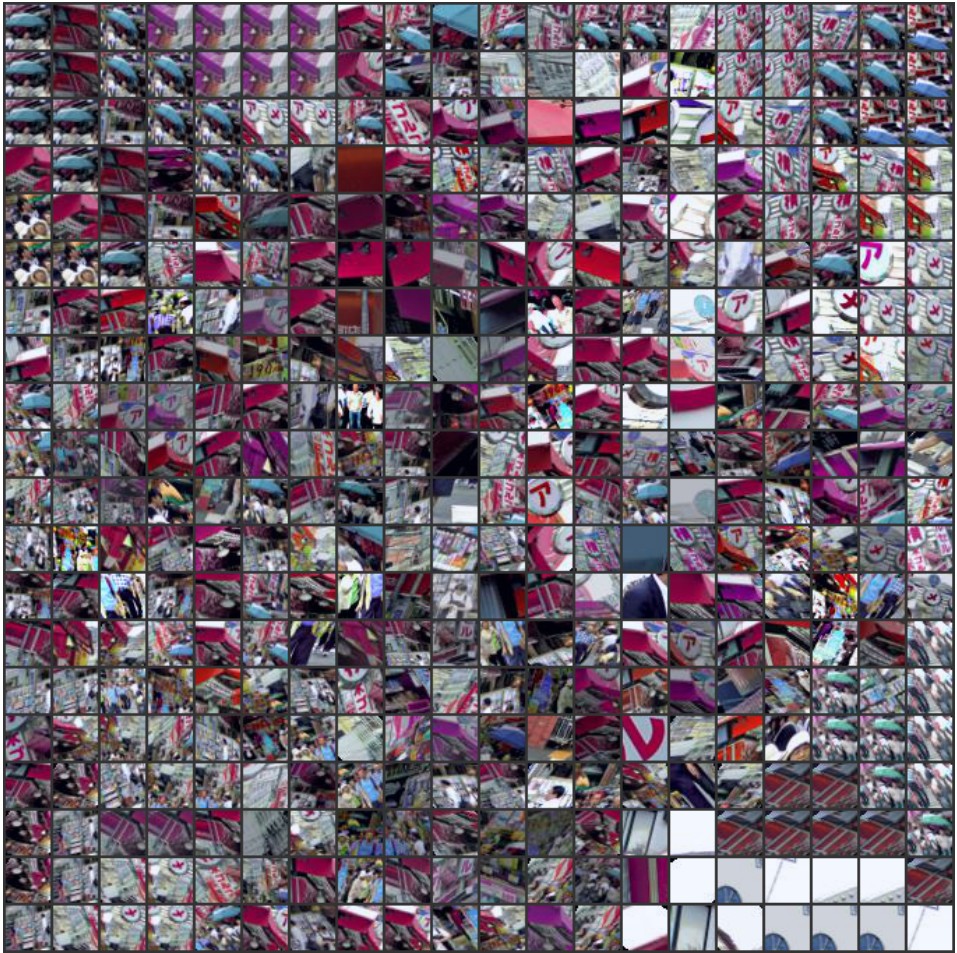

Figure 5: TSNE Manifold visualisation of distillation dataset corresponding to TSNE scatter plot in Fig. 3b).

# B FINDING OPTIMAL HYPER-PARAMETERS FOR SELECTION ALGORITHMS

## B.1 KMEANS BALANCING

To examine the impact of KMeans Balancing and its corresponding settings on the federated training, we conduct experiments while varying their values on multiple datasets having different numbers of prediction labels. The results have been presented in Table 8. Although values do not differ by a large margin for CIFAR10, CIFAR100 results provide us more assurance for the optimal values. We find that the KMeans selection strategy works best with a high number of clusters (K) compared to the number of classes in the corresponding classification problem (Table 8a), while forcing maximum class balancing while pruning (Table 8b) as well as selecting the hard examples (Table 8c) for distillation. Since the KMeans model is an independent model working on giving the pseudo-labels to the distillation training examples, there could be a play of correlation between the 3 hyper-parameters for this mechanism. It can be an interesting point for future research on this.

## B.2 ENTROPY SELECTION

For the entropy selection mechanism, we only have 2 settings to vary: Removal Percentage and Heuristics (Section 3.2). The results obtained during ablation studies with these settings have been presented in Table 9. From the results in Table 9a, it is clear that removing training examples with high confidence (less entropy) provides us best results. Removing a high percentage of training

| Number of Clusters | Private Dataset | |
|---|---|---|
| | CIFAR10 | CIFAR100 |
| 5 | $64.3 \pm 0.7$ | $19.9 \pm 1.9$ |
| 10 | $65.2 \pm 0.9$ | $21.2 \pm 0.1$ |
| 50 | $65.3 \pm 0.3$ | $22.9 \pm 0.9$ |
| 100 | $65.5 \pm 1.1$ | $18.9 \pm 2.9$ |
| 1000 | $\mathbf{66.3 \pm 1.1}$ | $\mathbf{22.9 \pm 0.1}$ |

(a) Varying cluster number with **selection heuristic = easy** and **balancing factor = 0.1**.

| Balancing Factor | Private Dataset | |
|---|---|---|
| | CIFAR10 | CIFAR100 |
| 0.0 | $64.9 \pm 1.3$ | $22.5 \pm 0.2$ |
| 0.05 | $65.0 \pm 1.7$ | $22.5 \pm 0.4$ |
| 0.1 | $65.2 \pm 2.0$ | $21.3 \pm 0.1$ |
| 0.5 | $\mathbf{66.6 \pm 1.4}$ | $22.2 \pm 0.1$ |
| 1.0 | $66.0 \pm 1.2$ | $\mathbf{24.0 \pm 1.4}$ |

(b) Varying balancing factor with **number of clusters (K) = 1000** and **selection heuristic = easy**.

| Selection Heuristics | Private Dataset | |
|---|---|---|
| | CIFAR10 | CIFAR100 |
| Easy | $65.8 \pm 1.3$ | $22.1 \pm 0.4$ |
| Hard | $\mathbf{66.7 \pm 1.1}$ | $\mathbf{23.8 \pm 1.8}$ |
| Mixed (50-50) | $65.5 \pm 2.1$ | $21.8 \pm 1.0$ |

(c) Varying selection heuristics with **number of clusters (K) = 1000** and **balancing factor = 1.0**.

Table 8: Best test set accuracy achieved during 30 rounds of training with KMeans balancing under different settings with different private datasets (Distribution $\alpha = 1.0$) using single image patches as the distillation dataset with 20% FedAvg initialisation rate on ResNet-8 (across 2 seeds).

examples from a large initial set using this mechanism also provided us with more robust training, compared to removing a smaller number of training examples from a small initial set (Table 9b). Similar to our last experiments with the KMeans mechanism, the results are more clearly pronounced in the presence of a more difficult dataset (100 classes compared to 10 in the case of CIFAR100 and CIFAR10).

| Removal Heuristic | Private. Dataset | |
|---|---|---|
| | CIFAR10 | CIFAR100 |
| Top | $67.0 \pm 0.9$ | $\mathbf{26.2 \pm 1.9}$ |
| Bottom | $61.9 \pm 2.4$ | $15.4 \pm 2.4$ |
| Random | $\mathbf{67.4 \pm 1.5}$ | $23.2 \pm 0.9$ |

(a) Varying Removal Heuristic with **Removal Percentage = 90%**.

| Removal Percentage (%) | Private Dataset | |
|---|---|---|
| | CIFAR10 | CIFAR100 |
| 10 | $65.0 \pm 0.9$ | $24.4 \pm 1.2$ |
| 50 | $\mathbf{67.5 \pm 1.2}$ | $25.8 \pm 1.4$ |
| 90 | $66.2 \pm 1.5$ | $\mathbf{26.7 \pm 1.5}$ |

(b) Varying Removal Percentage with **Removal Heuristic = Top**.

Table 9: Best test set accuracy achieved during 30 rounds of training with Entropy selection under different settings with different private datasets (Distribution $\alpha = 1.0$) using single image patches as the distillation dataset with 20% FedAvg initialisation rate on ResNet-8 (across 2 seeds).

## C  LEARNING RATE OPTIMISATION

### C.1  CLIENT SIDE LR

We find the optimal learning rate for local model training by using vanilla FedAvg as represented in Table 10 on ResNet-8. We do a grid search across a certain set of values for it, which is described in Table 10.

### C.2  SERVER SIDE LR

To find the optimal learning rate for the distillation training, we use FedDF as depicted in Table 11. This is accomplished with the help of a grid search across certain values for the distillation training learning rate, as mentioned in Table 11 itself.

| Local L.R. | Accuracy |
|---|---|
| 0.1 | 74.6 |
| 0.05 | 80.1 |
| 0.01 | **80.9** |
| 0.005 | 80.6 |
| 0.001 | 74.6 |

Table 10: Highest test accuracy achieved during 30 rounds of training with FedAvg (no distillation) on CIF10 with ResNet-8 (Distribution $\alpha = 100.0$).

| Global L.R. | Accuracy |
|---|---|
| 0.1 | $44.8 \pm 1.4$ |
| 0.05 | $62.7 \pm 1.0$ |
| 0.01 | $76.6 \pm 0.3$ |
| 0.005 | $\mathbf{78.1 \pm 0.4}$ |
| 0.001 | $75.2 \pm 1.0$ |

Table 11: Highest test accuracy achieved during 30 rounds of training with client learning rate = 0.01 using FedDF on CIF10 with ResNet-8 on 2 different seeds. (Distribution $\alpha = 100.0$)