# OpenReview forum: "Federated Learning with a Single Shared Image"
_ICLR.cc/2024/Conference — ICLR 2024 Conference Withdrawn Submission_

### Official Review · Reviewer_7jD4 · 2023-10-29

**Soundness:** 2 fair
**Presentation:** 1 poor
**Contribution:** 2 fair
**Rating:** 3
**Confidence:** 4

**Summary:**

This paper introduces a knowledge distillation approach for federated learning, in which a single shared image is required. Firstly, it utilizes the single image to generalize the distillation dataset by means of the transformation (e.g., rotation, flipping, and random crop). The authors propose patch subset selection (i.e., KMeans Balancing and Entropy Selection) to choose some samples from the dataset able to improve the global model's performance. The experiments include ablation studies and a comparison with FedDC. Ablation studies show how the single image, patch subset selection, and the neural network affect the performance of the proposed method, while the comparison depicts an improvement over FedDC.

**Strengths:**

1. It is interesting to apply knowledge distillation-based aggregation with a single image in federated learning.
2. The authors give a comprehensive discussion of the experiments, and the results seem reasonable and promising.

**Weaknesses:**

1. Followed by the first advantage, I think this work is a bit overclaimed. In my opinion, the size of a single shared image should be the same as the training images. However, in this paper, the shared image is of high resolution, which misleads the readers.
2. Followed by the first point, I am not convinced why a high-resolution image is more obtainable than a public dataset that contains the same size as the training data. According to Table 1, the high-resolution image cannot be randomly generated (e.g., random noise) and should align with the downstream task.
3. I think the presentation is incoherent. As mentioned in the introduction, the proposed work alleviates the heterogeneity of client architectures. However, it is unclear how the challenge is solved. According to your experimental settings, all clients' models are identical. Besides, Figure 1 shows a step in which the server broadcasts the global model to the clients.
4. The authors describe how to generate the distillation dataset. However, the paper lacks the crucial steps of federated learning (FL), i.e., how to leverage the dataset to train and aggregate the local models. Also, its contribution to FL is marginal because it largely follows the existing framework, according to Figure 1.

**Questions:**

In addition to the above weaknesses, I have one more question for the authors to clarify:

1. What does FedAvg initialization mean? Do you mean training the global model for multiple rounds with the conventional FedAvg and without knowledge distillation? I cannot see the definition of FedAvg initialization. The authors should include the details in the paper.

---

### Official Review · Reviewer_qTUJ · 2023-10-31

**Soundness:** 2 fair
**Presentation:** 2 fair
**Contribution:** 2 fair
**Rating:** 3
**Confidence:** 5

**Summary:**

Knowledge distillation (KD) in federated learning (FL) helps to transfer knowledge of local models to global models. However, existing KD methods require a shared dataset, which violates the privacy principle of FL. In order to mitigate this issue, the authors propose to generate multiple crops from each image as KD data. Moreover, they propose two dataset pruning strategies: 1) a kmeans-based and 2) an entropy-based method, to select the most informative crops for KD. Experiments on various datasets verify the effectiveness of the proposed method.

**Strengths:**

1.	It is impressive that only one image is needed to perform KD.
2.	The provided dataset pruning strategies are helpful.

**Weaknesses:**

1.	What would happen if we increase the number of KD images? I would appreciate it if the authors could provide more related ablation results.
2.	Comparisons against FedAvg and other federated distillation baselines are missing.
3.	Some data-free approaches such as (Zhu et al., 2021b) and “DENSE: Data-Free One-Shot Federated Learning” [NeurIPS 2022] that had completely removed any shared image between server and client, so what would be the unique advantages of using single images in this work? This work mentioned reducing the bandwidth costs, but did not report how much cost can be saved.
4.	Fonts in Figure 1 is too small.
5.	The chosen datasets are all simple ones, results on more complex dataset like Imagenet are expected.

**Questions:**

1.	What are the exact images mentioned in Table 1?
2.	How to make the proposed method be compatible with personalized FL?
3.	How the shared single image look like for each dataset?

---

### Official Review · Reviewer_9upr · 2023-10-31

**Soundness:** 1 poor
**Presentation:** 1 poor
**Contribution:** 2 fair
**Rating:** 3
**Confidence:** 4

**Summary:**

The paper proposes a new method to improve knowledge distillation under the federated learning (FL) framework. The method proposes to use only a single shared image between the client and the server to achieve knowledge distillation.

**Strengths:**

The knowledge distillation method under the FL framework is an interesting area for research, as it allows heterogeneous client model architecture to be able to aggregate at the central server.

**Weaknesses:**

- The writing in general requires significant improvement, with quite a lot of grammar mistakes and some confusing sentences.

- The main method of the paper is not presented well. Normally, in the method section (section 3), the authors should state the problem setups, the objective of the problem with clear definitions, etc. Also, it lacks detailed references; for example, if the patchification techniques are used previously in the KD methods, etc. Again, some notations in the 'entropy selection' and 'KMeans balancing' are not clearly defined, making the proposed method hard to follow.

- The paper doesn't mention where the single images come from. Is it under the assumption that there will be shared images saved on the server or how are the single images selected?

- For the FedDF method, the shared images for KD are only saved on the server side and it is unlabeled. But here in Figure 1, we can see that patches are sent from the server to the clients, but the paper doesn't state why the image needs to be shared with the client.

- Evaluation parts is lacking too: (1) need to be specific about how many clients are in the client pool, where the single image comes from, if the single image is randomly selected and if the single image impacts the performance; (2) only use ResNet based model, while FedDF (baselines) use more diverse model architecture; (3) FedAvg Initialisation rate requires definition; (3) FedDF is served as the baseline, but the paper doesn't mention the experimental setup for the baselines.

**Questions:**

See above in the 'weakness' section.

---

### Official Review · Reviewer_LPBt · 2023-10-31

**Soundness:** 3 good
**Presentation:** 3 good
**Contribution:** 3 good
**Rating:** 5
**Confidence:** 3

**Summary:**

In summary, the paper works on the problem of ensemble knowledge transfer to improve performance in federated learning while reducing the cost of public data transmission or data generation. Using augmented image patches generated from one image, they show that their method can improve the performance.

**Strengths:**

* Using dataset pruning and single data KD is new in federated learning.
* Authors show results with different model architectures and domain datasets, which is valuable and interesting.
* The evaluations show the practicality of the method for the target datasets.

**Weaknesses:**

* Authors should consider more recent baselines for KD-based FL methods.
* Could you please elaborate on how your method differs from synthetic data generation (by the server or clients) or dataset distillation in federated learning?
* Computation cost, especially for clients, is missing.

**Questions:**

* How would the total number of clients, local dataset size, and number of participants in each round affect the performance?
* How does the method work in highly non-iid settings?
* What is the computational overhead of your method?

---

### Author Response · Authors · 2023-11-19
**Thanks for your valuable feedback!**

Dear reviewers and the committee,

Thanks for presenting us with your valuable and critical feedback on our work by taking your time. This should surely help us refine our work in the future. For now, we have decided to withdraw our work from the current conference.

Kind Regards,
Sunny, Aaqib, and Yuki